# RNA polymerase backtracking results in the accumulation of fission yeast condensin at active genes

Julieta Rivosecchi*, Daniel Jost*, Laetitia Vachez, François DR Gautier, Pascal Bernard, Vincent Vanoosthuyse

The mechanisms leading to the accumulation of the SMC complexes condensins around specific transcription units remain unclear. Observations made in bacteria suggested that RNA polymerases (RNAPs) constitute an obstacle to SMC translocation, particularly when RNAP and SMC travel in opposite directions. Here we show in fission yeast that gene termini harbour intrinsic condensin-accumulating features whatever the orientation of transcription, which we attribute to the frequent backtracking of RNAP at gene ends. Consistent with this, to relocate backtracked RNAP2 from gene termini to gene bodies was sufficient to cancel the accumulation of condensin at gene ends and to redistribute it evenly within transcription units, indicating that RNAP backtracking may play a key role in positioning condensin. Formalization of this hypothesis in a mathematical model suggests that the inclusion of a sub-population of RNAP with longer dwell-times is essential to fully recapitulate the distribution profiles of condensin around active genes. Taken together, our data strengthen the idea that dense arrays of proteins tightly bound to DNA alter the distribution of condensin on chromosomes.

## Introduction

Structural maintenance of chromosomes (SMC) complexes are essential for the organization and stability of chromosomes from bacteria to humans (Uhlmann, 2016; Hassler et al, 2018; van Ruiten & Rowland, 2018). The SMC complex condensin is particularly important for the compaction and the structuration of chromosomes throughout mitosis and for their faithful segregation to daughter cells (Hirano, 2016). Condensin is a ring-shaped DNA translocase that uses the energy of ATP-hydrolysis to organize mitotic chromosomes into large consecutive loops of chromatin (Naumova et al, 2013; Gibcus et al, 2018). It has been shown in vitro that purified condensin hydrolyses ATP to extrude loops of naked DNA (Ganji et al, 2018; Kong et al, 2020), but the structural details of the formation and enlargement of such loops remain poorly understood (Cutts & Vannini, 2020). Furthermore, whether such loop extrusion activity is the only way that condensin complexes organize mitotic chromosomes is still under debate as condensins and other architectural proteins may also participate in the organization of chromosomes by bridging-induced phase separation (Cheng et al, 2015; Sakai et al, 2018; Ryu et al, 2021).

Another fundamental question is to understand how chromatin and large DNA-bound protein assemblies impact the loop extrusion activity of condensin in vivo. Loop extrusion on chromatin in vivo is predicted to be roughly 10 times slower than on naked DNA in vitro (Banigan & Mirny, 2020), and it was recently suggested that arrays of proteins tightly bound to DNA could hinder the loop extrusion activity of condensin, possibly by constituting a steric obstacle to the reeling of chromatin (Guérin et al, 2019). An inability to bypass obstacles might result in the formation of unlooped chromatin gaps within mitotic chromosomes (Banigan et al, 2020). Whether and how condensin bypasses chromatin-associated obstacles is currently unclear.

Gene transcription has been shown to influence the distribution of SMC complexes in several organisms. In *Bacillus subtilis* and *Caulobacter crescentus*, a single SMC complex juxtaposes the arms of a circular chromosome by translocating in a unidirectional fashion from a single loading site (Gruber & Errington, 2009; Sullivan et al, 2009; Le et al, 2013; Wang et al, 2015; Tran et al, 2017). A highly active transcription unit in the opposite direction (head-on orientation) was shown to slow down the translocation of SMC, which transiently accumulates towards the 3' of the unit in a transcription-dependent manner (Tran et al, 2017; Wang et al, 2017; Brandão et al, 2019). It has been proposed that RNA polymerase (RNAP) molecules themselves constitute a directional albeit permeable barrier that impedes the translocation of SMC and each encounter with a RNAP molecule would force SMC to stall for a few seconds (Brandão et al, 2019). Great densities of RNAP (or other DNA-bound proteins) are therefore expected to impact the distribution of SMC along chromosome arms. Interestingly, specific mutations in *B. subtilis* SMC were shown to interfere with its ability to overcome transcription-dependent obstacles, suggesting that

Laboratoire de Biologie et Modélisation de la Cellule, Université de Lyon, École Normale Supérieure de Lyon, Centre National de la Recherche Scientifique (CNRS), Unité Mixte de Recherche (UMR) 5239, Lyon, France

Correspondence: vincent.vanoosthuyse@ens-lyon.fr
Julieta Rivosecchi's present address is Department of Cellular, Computational and Integrative Biology–CIBIO, University of Trento, Trento, Italy
*Julieta Rivosecchi and Daniel Jost contributed equally to this work

their bypass is an active process (Vazquez Nunez et al, 2019). Consistent with this, it was postulated that the bypass rate of different SMC complexes is a function of their intrinsic ATP hydrolysis rates (Brandão et al, 2019). In eukaryotes, transcription is also a positioning device for the SMC complex cohesin in interphase (Lengronne et al, 2004; Bausch et al, 2007; Busslinger et al, 2017; Heinz et al, 2018), suggesting that transcription is a conserved regulator of SMC occupancy.

Transcription also impinges on the distribution of condensin complexes in eukaryotes, even in organisms where active transcription is strongly reduced in mitosis when the association of condensin with chromosomes is the strongest (Bernard & Vanoosthuyse, 2015). In both chicken and human cells, condensin I, which only associates with predominantly transcriptionally silent chromatin after nuclear envelope breakdown in mitosis, accumulates towards the 5′ of RNAP2-transcribed genes that were highly transcribed in the previous G2 phase (Kim et al, 2013; Sutani et al, 2015). Similarly, in mouse ES cells and human cells, the localisation of condensin II, which is nuclear throughout the cell cycle, correlates with RNA polymerase II (RNAP2) occupancy, and human condensin II accumulates at the 3′ end of highly transcribed genes in interphase cells (Dowen et al, 2013; Iwasaki et al, 2019). By contrast, RNA polymerase I (RNAP1) transcription was proposed to antagonize the accumulation of condensin within the 35S transcription unit in budding yeast (Johzuka & Horiuchi, 2007; Clemente-Blanco et al, 2009). The role of transcription in establishing condensin-accumulating regions is therefore unclear. Fission yeast is a very good model to understand how transcription affects condensin because transcription remains active during mitosis, when the activity of condensin is maximal. A number of studies have established that fission yeast condensin accumulates in a transcription-dependent manner in the vicinity of genes that are highly expressed in mitosis, whatever the RNAP involved (RNAP1, RNAP2, or RNAP3) (Nakazawa et al, 2008, 2015; Kim et al, 2014, 2016; Sutani et al, 2015). Moreover, the drug-induced inhibition of transcription partially rescued the loss of viability of condensin-defective mutants (Sutani et al, 2015) and it was recently proposed that active transcription interferes locally with the condensin-dependent resolution of sister chromatids (Nakazawa et al, 2019b Preprint). Taken together, these observations suggest that transcriptionally active RNAPs, and/or features associated with ongoing transcription, might challenge condensin function and the assembly of mitotic chromosomes in fission yeast.

We have previously proposed that fission yeast condensin might load onto DNA at nucleosome-depleted promoters of active genes (Toselli-Mollereau et al, 2016) and would subsequently accumulate particularly towards the 3′ of genes actively transcribed by RNAP2 (Sutani et al, 2015; Toselli-Mollereau et al, 2016). Considering the relatively small size of transcription units in fission yeast (~2 kb for protein-coding genes), a condensin complex loaded at the promoter might be more likely to reach the 3′ of genes in a head-to-tail than in a head-to-head orientation. It is therefore unclear whether the accumulation of condensin in the 3′ of genes is due to a head-on conflict between transcription and translocating condensin, like in bacteria (see above). On the other hand, there is evidence that the positioning of fission yeast condensin at the 3′ of RNAP2-transcribed genes could be functionally linked to the process of transcription termination. First, a number of positive and negative genetic interactions have been reported between mutants of the

transcription termination machinery and mutants of condensin (Vanoosthuyse et al, 2014; Nakazawa et al, 2019a). As lack of condensin does not directly impact transcription termination in fission yeast (Hocquet et al, 2018; Nakazawa et al, 2019a), these genetic interactions suggest that RNAP2 transcription termination mechanisms might impinge on the function of condensin. Consistent with this interpretation, it was shown recently that to inactivate Xrn2[Dhp1], an enzyme that is key for RNAP2 transcription termination, was sufficient to displace condensin further downstream of active transcription units (Nakazawa et al, 2019a), strengthening the possibility of interplay between transcription termination mechanisms, the 3′ edge of the RNAP2 domain and the positioning of condensin. To explain these observations, it was proposed that condensin is actively recruited at transcription termination regions because they accumulate single-stranded DNA and/or chromatin-associated RNA molecules that interfere with the organization of mitotic chromosomes (Sutani et al, 2015; Nakazawa et al, 2019a). Condensin, thanks to its ability to re-anneal melted dsDNA molecules in vitro (Sutani & Yanagida, 1997; Sakai et al, 2003; Akai et al, 2011; Sutani et al, 2015), would suppress these structures, thereby allowing the formation of fully functional mitotic chromosomes. This hypothesis therefore posits that condensin plays a "clearing" role in the assembly of mitotic chromosomes (Yanagida, 2009) besides its role in the extrusion of chromatin loops. It remains unclear, however, how short chromosome regions that are rich in single-stranded DNA and/or chromatin-associated RNA could interfere with the formation of segregation-competent mitotic chromosomes. Importantly, other models could also account for these observations: (i) a permeable moving barrier model as described in bacteria (Brandão et al, 2019) could explain the accumulation of translocating condensin at the 3′ border of the RNAP2 domain or (ii) the transcription termination machinery could play a more direct role in the positioning of moving condensin. These models have not yet been tested experimentally.

Here we sought to better understand what features of transcription might influence the distribution of condensin in fission yeast mitosis. By switching the orientation of an RNAP2-transcribed gene expressed in mitosis, we tested whether or not gene transcription could be a directional barrier for condensin. Although these experiments neither confirmed nor infirmed that transcription might be a directional barrier in fission yeast, they strongly reinforced the idea that the 3′ end of genes contain intrinsic condensin-positioning features. We then showed that to interfere with RNAP3 transcription termination also alters the distribution of condensin, suggesting that transcription termination defects impact the accumulation of condensin, whatever the RNAP involved. This strengthened the idea that RNAP molecules rather than a specific transcription termination machinery could influence the positioning of condensin. Consistent with this, we provide evidence that to increase the stability of backtracked RNAP2 polymerases throughout the gene body was sufficient to shift condensin occupancy towards the 5′ end of transcribed genes. This strongly suggests that backtracked RNAP molecules are themselves positioning devices for condensin. We used mathematical modelling to formalize this hypothesis and determined that simulations that take into account the presence of two distinct RNAP populations, one mobile and one backtracked, more closely predict the distribution

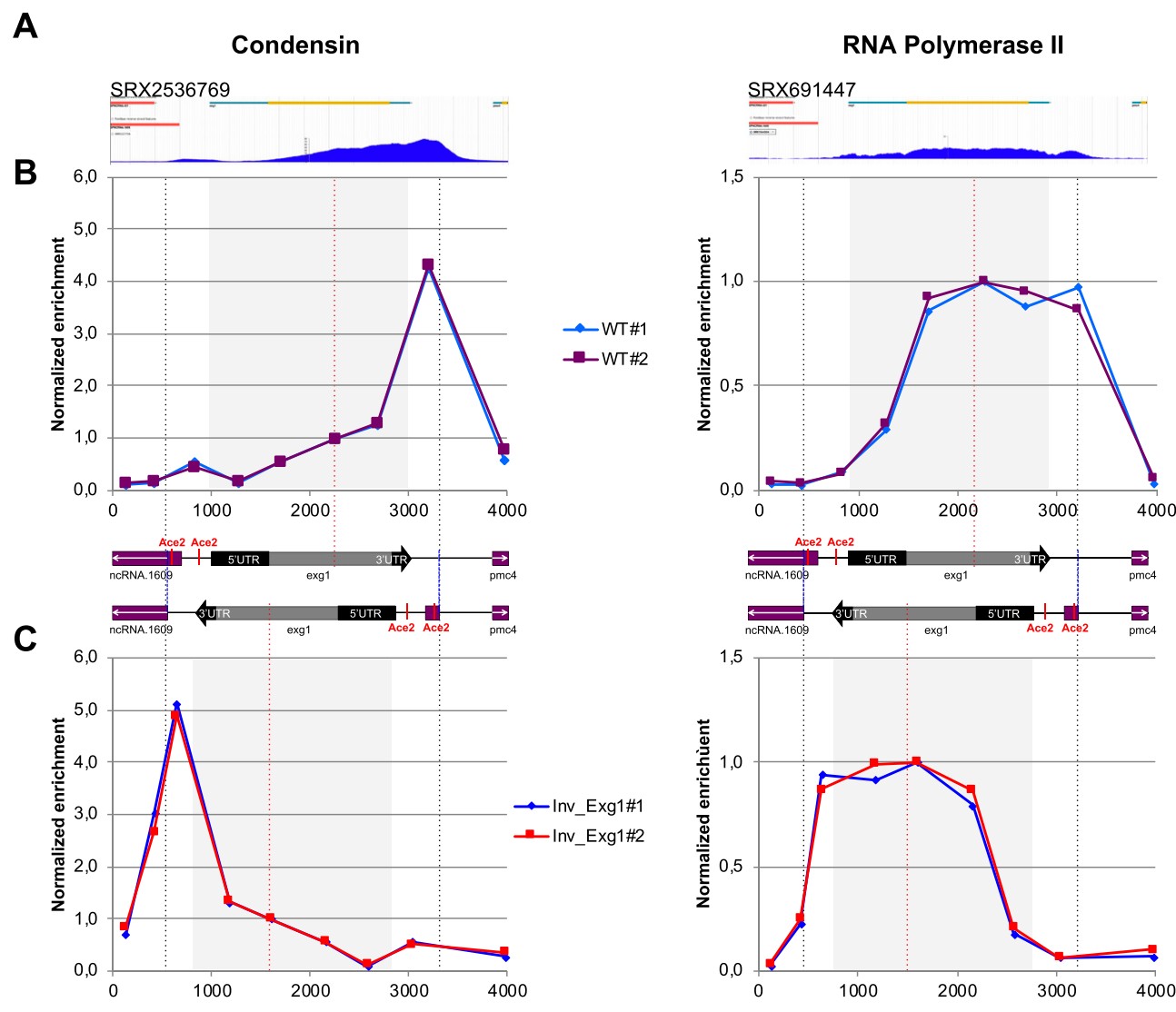

**Figure 1. Distribution of condensin and RNA polymerase 2 upon flipping of *exg1*.**
**(A)** ChIP-seq profiles of condensin (left) and the RNA polymerase subunit Rpb5 (right) around *exg1* in mitotic cells. The ChIP-seq data are indicated by their Sequence Read Archive numbers and were generated in Sutani et al (2015) and Kakui et al (2017), respectively. **(B)** Cells were synchronized in mitosis and ChIP-qPCR in two different biological replicates was used to determine the distribution of condensin (left) and Rpb1 (right) around *exg1*. **(C)** Same as (B) when the orientation of *exg1* has been flipped over. The scheme above shows the organization of the chromosome around *exg1* in the wild-type (top) and in the reversed (bottom) orientations. Vertical dotted lines indicate the region of the chromosome that has been flipped over. Grey squares indicate the position of the *exg1* transcription unit. The % IP were normalized using the values given at the site within the gene body indicated by the red vertical dotted line (*exg1#1*). The raw data are shown in the source data files.
Source data are available for this figure.

pattern of condensin around active genes in fission yeast than those that only consider the mobile population of RNAP, as previously described (Brandão et al, 2019). Taken together, our data clarify the role of transcription in the accumulation of condensin and are consistent with the idea that proteins that are tightly bound to DNA impact the distribution of condensin along mitotic chromosomes.

## Results

One prediction of the permeable moving barrier model is that the orientation of transcription impacts the distribution pattern of condensin (Brandão et al, 2019). To test this prediction in fission yeast, we changed the orientation of *exg1*, a gene that is transcribed by RNAP2 in mitosis and where condensin was shown previously to accumulate strongly towards the termination zone (Kakui et al, 2017) (Fig 1A). Interestingly, it was shown that RNAP2 levels remain relatively constant throughout the gene (Sutani et al, 2015) (Fig 1A), arguing that the density of RNAP2 per se is unlikely to account for the position of condensin at the 3′ end of this gene. Importantly, the reversal of orientation did not affect RNAP2 levels around *exg1* in mitotic cells (Fig 1B and C). Strikingly, the peak of condensin accumulation was moved symmetrically with the flipping of *exg1* and coincided with the new genomic position of the 3′ end of the gene (Fig 1B and C). These observations could be interpreted in several ways: either (i) transcription is not a directional barrier

for condensin in fission yeast, or (ii) transcription is a directional barrier for condensin but the chromatin around *exg1* can be reeled by condensin from both directions with equal probability; alternatively, (iii) the transcription termination process itself or its machinery forces the accumulation of condensin in the 3′ end of transcribed genes.

To test the latter hypothesis, we assessed whether transcription termination at another class of genes also modulates the distribution of condensin. Several ChIP-seq studies reported that fission yeast condensin accumulates at RNAP3-transcribed genes (Kim et al, 2014, 2016; Sutani et al, 2015; Kakui et al, 2017) and it was proposed that the B-box binding transcription factor TFIIIC and the TATA-binding protein Tbp1 were required for this accumulation by interacting directly with condensin (Iwasaki et al, 2010, 2015). Whether or not transcription termination at RNAP3-transcribed genes could impact the distribution of condensin was not investigated. We recently demonstrated that the conserved DNA & RNA helicase Sen1 is required for efficient transcription termination at RNAP3-transcribed genes in -*cis* (Rivosecchi et al, 2019). In the absence of Sen1, RNAP3 strongly accumulates downstream of most of its target genes and we showed that this accumulation of read-through RNAP3 molecules downstream of gene ends could be suppressed by strengthening the endogenous terminators by the use of long polyT sequences (Rivosecchi et al, 2019). We tested whether the RNAP3 termination defects associated with lack of Sen1 could impact the distribution of condensin around RNAP3-transcribed genes. Strikingly, condensin levels increased significantly at a subset of RNAP3-transcribed genes in synchronized mitotic cells lacking Sen1 (Fig 2A). This accumulation was specific because lack of Sen1 had no impact on the association of the heterologous *Escherichia coli* protein LacI expressed in fission yeast cells (Fig 2A). Importantly, the accumulation of condensin in *sen1Δ* cells could not be caused by an accumulation of either TFIIIC or Tbp1 because their levels on chromatin remained largely unaffected in the absence of Sen1, as shown by ChIP with a GFP-tagged version of Tbp1 and a myc-tagged version of the TFIIIC component Sfc6 (Figs 2B and S1). In the absence of Sen1, condensin did not accumulate either at chromosome-organizing clamps sites (Fig 2A), which recruit TFIIIC but not RNAP3 (Noma et al, 2006), consistent with a transcription-mediated effect. To further determine whether the accumulation of condensin was mechanistically linked to the transcription termination defects observed in the absence of Sen1, we corrected those defects by strengthening the terminator sequences at two tRNA genes by inserting long polyT sequences, as described previously (Rivosecchi et al, 2019). As expected, this strategy was sufficient to correct the accumulation of RNAP3 downstream of the terminator sequences in mitotic cells lacking Sen1 (Fig 2C and D, lower panels). Strikingly, this was also sufficient to prevent the accumulation of condensin (Fig 2C and D, top panels). These observations show that the increased accumulation of condensin at class III genes in the absence of Sen1 is a direct consequence of RNAP3 transcription termination defects. This is reminiscent of the data showing that to interfere with RNAP2 transcription termination mechanisms also altered the distribution of condensin (Nakazawa et al, 2019a) or cohesin (Heinz et al, 2018) in -*cis*. As the transcription termination machineries differ for RNAP2 and RNAP3, it seems unlikely that a component of the transcription

termination machinery itself is involved in the positioning of condensin in the 3′ of genes. Our data suggest instead that intrinsic properties of RNAP molecules undergoing a termination process might explain their impact on the distribution of condensin.

What could be the intrinsic properties of RNAP molecules in the 3′ end of genes that impact the position of condensin? We hypothesized that RNAP backtracking could be a contributing factor for two reasons: (i) RNAP molecules are often backtracked around termination sites (Lemay et al, 2014; Sheridan et al, 2019) and (ii) backtracking would conceivably strengthen the interaction of RNAP molecules with chromatin, making them less dynamic and possibly a harder obstacle to bypass by translocating condensin molecules (Brandão et al, 2019; Guérin et al, 2019). To test this hypothesis, we sought to prolong RNAP2 backtracking events by over-expressing a dominant-negative mutant of TFIIS (*tfs1D274AE275A* in fission yeast [Lemay et al, 2014], thereafter referred to as *tfs1DN*). This strategy was shown to interfere with transcription elongation throughout the gene in different organisms and to alter the distribution of RNAP2 (Sigurdsson et al, 2010; Sheridan et al, 2019; Zatreanu et al, 2019). Upon *tfs1DN* expression in mitotic fission yeast cells, the distribution of RNAP2 was reduced in the 3′ and shifted towards the 5′ of genes (Fig 3). Remarkably, the over-expression of *tfs1DN* had a similar impact on the distribution of condensin around RNAP2-transcribed genes in mitosis (Fig 3): the accumulation of condensin at the 3′ of genes was significantly reduced but its accumulation towards the 5′ increased significantly. Overall, condensin became evenly distributed throughout the gene body upon *tfs1DN* over-expression instead of being enriched at the transcription termination site. On the contrary, over-expression of *tfs1DN* had no impact on the association of condensin with the RNAP1-transcribed 18S (Fig S2). Taken together, these observations are consistent with the idea that RNAP backtracking impacts the distribution of condensin within transcribed genes. Because backtracking is a prominent feature in the 3′ end of genes (Lemay et al, 2014; Sheridan et al, 2019), this might explain why condensin accumulates particularly over the 3′ of transcriptionally active genes in fission yeast (Sutani et al, 2015; Toselli-Mollereau et al, 2016).

To challenge this hypothesis, we used mathematical modelling of the interplay between condensin and RNAP (Figs 4, S3, and S4 and Supplemental Data 1). Previous models assumed that RNAP could push condensin towards the 3′ of the gene and that condensin could bypass RNAP after a small delay (Brandão et al, 2019). We first implemented this model (Fig 4A) and confirmed that it predicts an accumulation of condensin in the 3′ of genes if RNAP and condensin travel in opposite directions (Fig 4B and C), as reported previously (Brandão et al, 2019). Importantly, this predicted accumulation followed an almost exponential profile through the gene (Fig 4C and Brandão et al, 2019), which does not correspond to what is observed in vivo for fission yeast condensin or for human condensin II, where the accumulation of condensin increases gradually and slowly in the gene body with an additional strong peak around the termination zone (Figs 1 and 3 and Sutani et al, 2015; Iwasaki et al, 2019). We conclude that this first model cannot fully recapitulate observations made in vivo.

Importantly, this first model considered that all RNAP molecules have the same properties and the same dynamics throughout the transcription unit. This, however, does not correspond to the reality

**A** Condensin and LacI

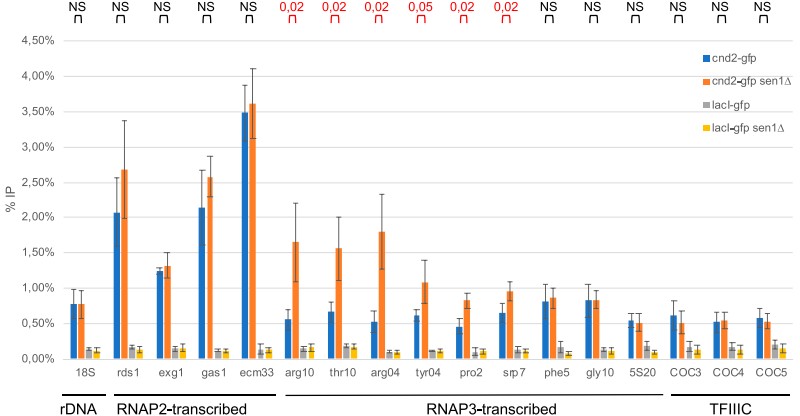

**B** Sfc6

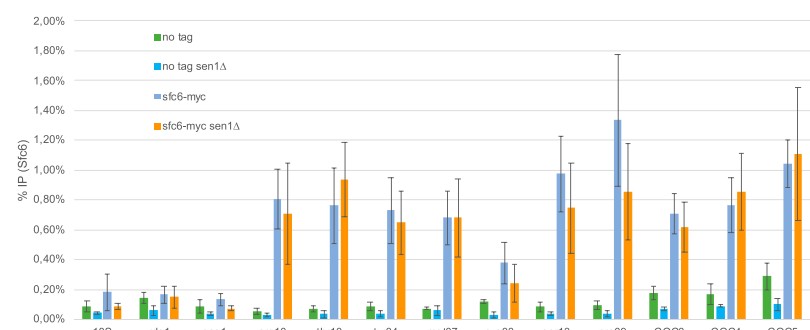

**C** Condensin   **D** Condensin

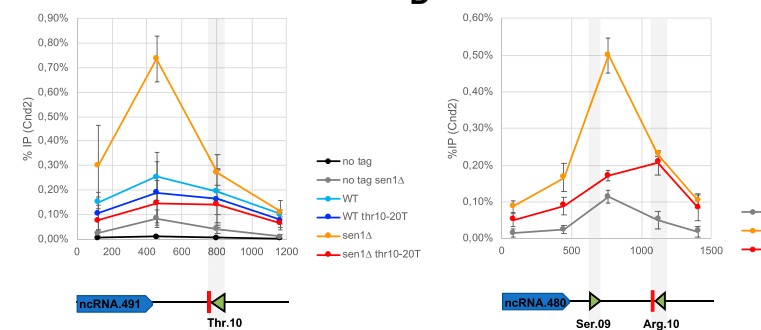

RNAP3   RNAP3

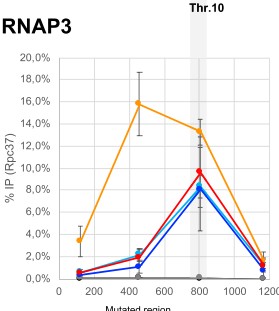
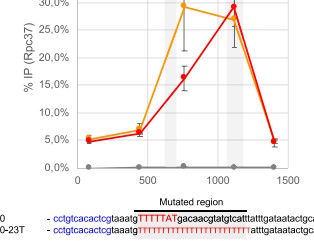

**Figure 2. RNA polymerase 3 transcription defects induced by lack of Sen1 trigger the accumulation of condensin.**
**(A)** Cells were synchronized in metaphase and the association of condensin (Cnd2-GFP) or the heterologous LacI (lacI-GFP) at the indicated loci was investigated by ChIP-qPCR in the presence and in the absence of Sen1 (mean ± std of four biological replicates; *P*-values determined by the test of Wilcoxon Mann-Whitney are indicated above the graph). **(B)** The association of the TFIIIC component Sfc6 at the indicated loci was investigated by ChIP-qPCR in cells synchronized in metaphase (mean ± std of five biological replicates). **(C, D)** Distribution of condensin (cnd2-GFP, top) and RNA polymerase 3 (rpc37-flag, bottom) around *SPCTRNATHR.10* (C) and *SPCTRNAARG.10* (D) in mitotic cells, in the presence or not of super-terminator sequences (*thr10-20T* and *arg10-23T*, respectively) which correct the transcription termination defects in the absence of Sen1 (Rivosecchi et al, 2019) (compare the yellow and red curves). **(C, D)** Results are presented as (mean ± std) of three (C) or four (D) biological replicates.
Source data are available for this figure.

of the transcription cycle, where RNAP frequently pauses and backtracks, notably in the termination zone (reviewed in Noe Gonzalez et al [2021]). We therefore built a second model (Fig 4D), where RNAP may dynamically switch between an elongating,

mobile form (in orange on Fig 4D) and a backtracked, immobile state (in red on Fig 4D). As in the previous model (Brandão et al, 2019), we considered that mobile RNAP molecules can push condensin towards the 3′ of the gene. In addition, we postulated that

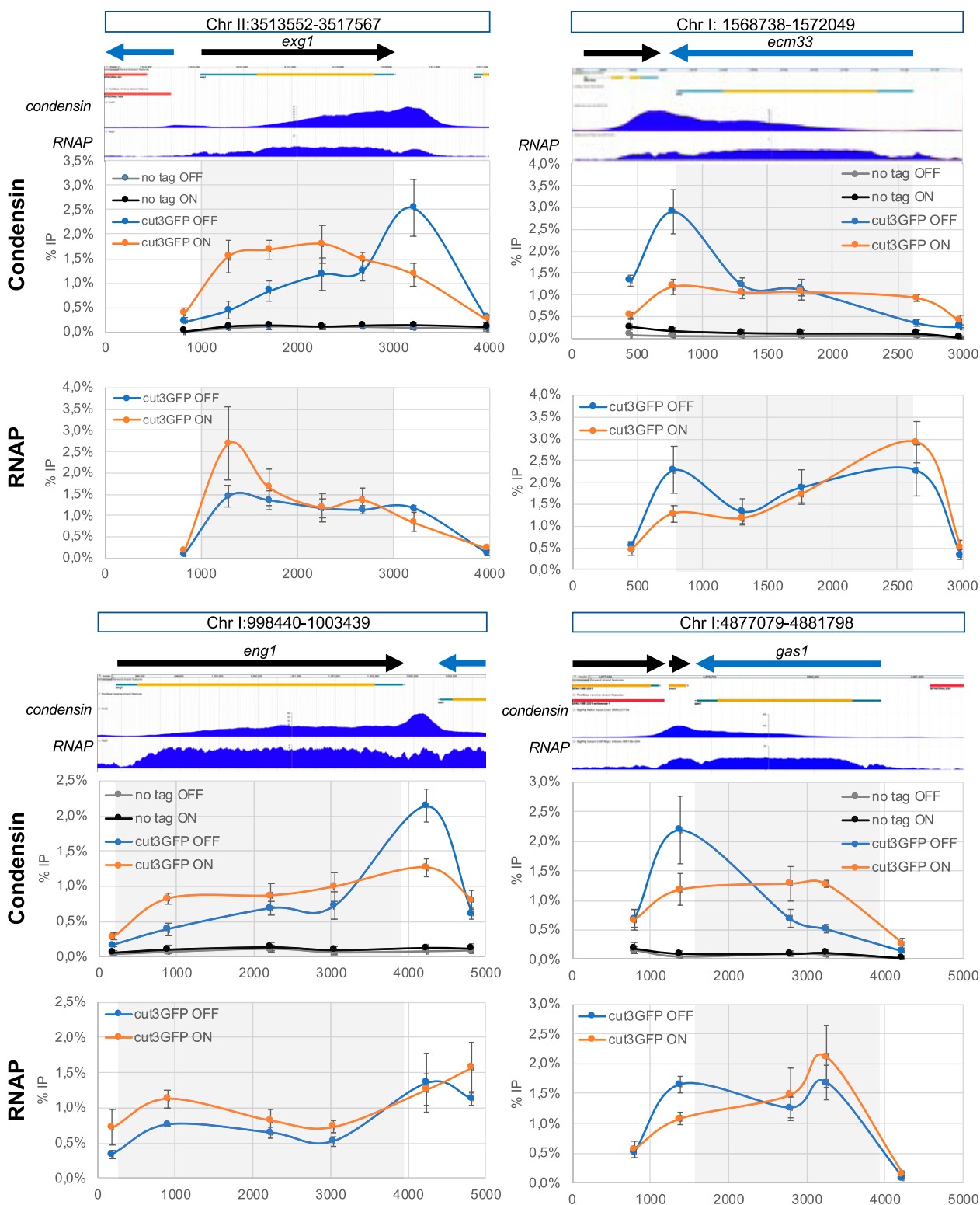

**Figure 3. The over-expression of *tfs1DN* alters significantly the distribution of condensin around RNA polymerase 2-transcribed genes.**
Cells were synchronized in metaphase and the association of condensin (cnd2-GFP) at the indicated loci was investigated by ChIP-qPCR (mean ± std of three biological replicates). Cells carried a plasmid allowing the AhTET-induced over-expression of *tfs1-DN*, as described previously (Lemay et al, 2014). DMSO was used as control. For each locus investigated, the normal distribution of condensin and RNA polymerase 2 as determined by ChIP-seq is shown above, as published in Sutani et al (2015) and Kakui et al (2017), respectively. The raw data are shown in the source data files.
Source data are available for this figure.

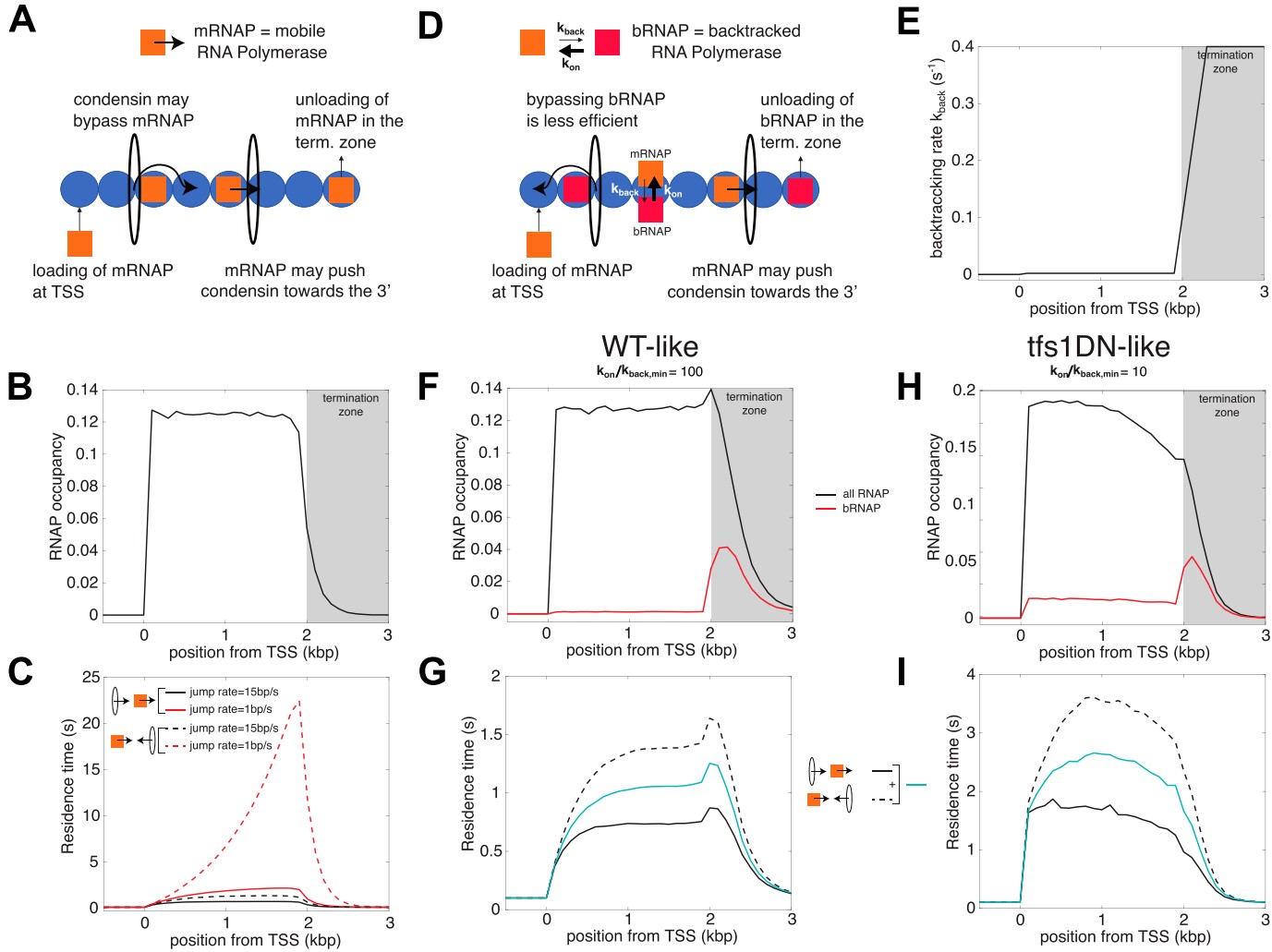

**Figure 4. Mathematical models formalize the role of RNA polymerase (RNAP) backtracking in the specific accumulation of condensin in the termination zone of active genes.**
**(A)** Interplay between the translocation of condensin and transcription—simple model (Brandão et al, 2019). Condensin translocates along chromatin from either 5′ to 3′ or 3′ to 5′. RNAPs bind to TSS, translocate unidirectionally from 5′ to 3′ and unbind when they reach the termination zone. The speed at which condensin translocates is reduced when it encounters a RNAP. Moving RNAPs can push condensin towards the 3′ of the gene if they represent an obstacle for their translocation (see the Materials and Methods section and Supplemental Data 1). **(B)** Profile of RNAP mimicking a typical WT-situation of a ~2 kbp-long gene (see e.g., Figs 1 and 3). **(C)** Residence time profiles of condensin along the gene when condensin and RNAP move in the same (head-to-tail, full lines) or opposite (head-to-head, dashed lines) direction for two different bypassing rates. **(D)** Interplay between the translocation of condensin and transcription—backtrack model. **(A)** This model makes the same basic assumptions as in (A) but RNAP can now dynamically switch between two states: either mobile (mRNAP) or backtracked (bRNAP). The reduction in condensin speed due to collisions with RNAP is stronger with bRNAP than with mRNAP (see the Materials and Methods section and Supplemental Data 1). **(E)** Rate of backtracking along the gene used in the model. **(F, G)** Wild-type situation. **(F)** Density of total RNAP (black line) and bRNAPs (red line) over a ~2 kbp-long gene. **(G)** Residence time profiles of condensin along the gene for head-to-tail (full black line) or head-to-head (dashed black line) collisions for a fast bypass rate over mobile RNAP (15 bp/s). The blue full line represents the average between both profiles. **(H, I)** Over-expression of *tfs1DN*. **(H)** Profiles of RNAP (black line) and backtracked RNAPs (red line) obtained by increasing the dwell-time of the backtracked state by 10-fold to mimic the *tfs1DN* situation. **(I)** As in (G) but for the *tfs1DN-like* simulations.

bypassing backtracked RNAPs is less efficient (Fig 4D). In normal conditions, the major site of RNAP backtracking would be the termination zone (Fig 4E and F). Importantly, as is observed in vivo, this second model predicted a gradual accumulation of condensin in the body of genes and a more pronounced accumulation in the termination zone (Fig 4G). Strikingly, the model made those predictions whatever the direction of travel of condensin (co-directional, full black line, or head-on, dotted black line, Fig 4G). This is reminiscent of our observations that

flipping *exg1* did not fundamentally change the distribution pattern of condensin (Fig 1).

We tested our model to see whether it could predict the changes to the distribution of condensin triggered by the over-expression of *tfs1DN* (Fig 3). In this situation, we assumed that the dwell-time of the backtracked state is strongly increased. As a result, the proportion of backtracked RNAP significantly increases inside the gene body and the global RNAP occupancy is shifted towards the 5′ of the gene (Fig 4H). Strikingly, our new model predicted that condensin

would accumulate more evenly throughout the gene body in those conditions and lose its specific accumulation in the termination zone (Fig 4I). This is in perfect agreement with our observations in vivo (Fig 3). We conclude that accounting for the presence of backtracked RNAP along the gene is a key ingredient to describe the pattern of condensin around active genes.

# Discussion

Taken together, our results strongly suggest that RNAP backtracking results in the accumulation of condensin in -cis. This could explain why both yeast condensin (this study and D'Ambrosio et al, 2008; Sutani et al, 2015) and human condensin II (Iwasaki et al, 2019) tend to accumulate in the 3' of active genes. Our data also confirm that to interfere with transcription elongation affects the distribution of condensin within genes (Fig 3), as predicted previously (Brandão et al, 2019). In the future, single molecule approaches could be used to determine experimentally whether condensin pauses for longer when facing backtracked RNAP molecules than when facing elongating, dynamic RNAP.

In the absence of Sen1, RNAP3 accumulates strongly over and downstream of class III genes but this accumulation is associated with reduced rates of transcription (Rivosecchi et al, 2019). We, therefore, speculate that those accumulated RNAP3 molecules are often backtracked and that the increased levels of condensin downstream of class III genes in the absence of Sen1 depend on the size and the density of the domain occupied by these read-through, backtracked polymerases. By introducing super-terminator sequences (Fig 2), we reduced the size of this domain and prevented the accumulation of condensin. Both the size and the density of this RNAP3-rich read-through domain would depend on the chromatin context and the transcription rate. This might explain why condensin did not accumulate at all RNAP3-transcribed genes in the absence of Sen1 (Fig 2). Note also that in our hands, RNAP3-transcribed genes are not strong condensin-accumulation sites when Sen1 is present (Fig 2) and we predict that this is because the size of the domain occupied by RNAP3 and TFIIIC is not large enough to constitute a significant obstacle (~100 bp).

Our data are also consistent with the observation that the number and density of DNA-bound Rap1 proteins influence condensin function in budding yeast (Guérin et al, 2019) and strengthen the idea that arrays of proteins that are tightly bound to DNA could trigger the accumulation of condensin. Similarly, it is conceivable that a number of tightly bound proteins (e.g., transcription factors or paused RNAP2 molecules) contribute to position condensin I in the 5' of genes in mitosis in vertebrates (Kim et al, 2013; Sutani et al, 2015), even in the absence of significant transcriptional activity. We predict that the same rules are likely to apply to other SMC complexes.

It remains to be determined why tightly bound proteins lead to the accumulation of condensin. They might represent regions where condensin is preferentially loaded but we favor the idea that they might oppose steric hindrance to the translocation of condensin, as argued previously (Guérin et al, 2019). Alternatively, they might modify the local physical properties of the chromatin fibre or

some properties of condensin itself in a way that would eventually challenge its translocation. Future work is needed to answer these important questions and to understand better how condensin and other SMC complexes work in the context of chromatin.

# Materials and Methods

### Yeast strains

The strains used in this study are listed in Table S1.

### Cell synchronization

Two different methods were used to synchronize fission yeast cells in metaphase. The first method (Figs 1B and C and 2C and D) used an analogue-sensitive version of the Cyclin-DK Cdc2 (cdc2-asM17 [Aoi et al, 2014]) which can be inhibited by 2 $\mu$M of 3-Br-PP1 (A602985; Toronto Research Chemicals). After 3 h in the presence of the drug at 28°C in rich medium, $5 \times 10^8$ cells were filtered, washed three times with warm medium and released in fresh medium without BrPP1. After 10 min, ~80% of cells were in mitosis, as judged by the localisation of GFP-tagged condensin (Cnd2) in the nucleus. The second synchronization method (Figs 2A and B and 3) relies on the inhibition of the expression of Slp1 (Petrova et al, 2013), a protein that is key to the metaphase to anaphase transition (Matsumoto, 1997). Cells expressing Slp1 under the control of the thiamine-repressible nmt41 promoter were grown in minimal medium at 32°C until mid-log phase, when 60 $\mu$M of thiamine was added to the culture for 3 h. Cell synchrony in mitosis was checked as above by the presence of GFP-tagged condensin (Cnd2) in the nucleus.

### Exg1 inversion

ura4 was first integrated at the exg1 locus to generate the exg1Δ::ura4+ strain. PCR was then used to fuse the 3' of exg1 to its 5' domain and its 5' to its 3' domain using the primers exg1 qL2/exg1 RV3 and exg1 qR2/exg1 FW3 (see Table S2 for a list of the primers used in this study). An overlapping PCR was then used to amplify the whole inverted locus. The resulting 2.8 kb PCR product was then transformed into the exg1Δ::ura4+ strain and stable integrants were selected by several rounds of 5-fluoroorotic acid (FOA) selection. The correct integration of the exg1 gene in the reverse orientation was confirmed by PCR and sequencing.

### tfs1DN over-expression

The strains of interest were transformed with the pFB818 plasmid (a generous gift from François Bachand, University of Sherbrooke) that allows the inducible expression of tfs1-DN by addition of 7.5 $\mu$M of anhydrotetracycline hydrochloride (AhTET, 94664; Sigma-Aldrich), as described in Lemay et al (2014). AhTET was dissolved in DMSO. Cells were grown in PMG-Leu at 30°C until they reached a concentration of $5 \times 10^6$ cells/ml. AhTET or DMSO was added for 3 h, at which point 60 $\mu$M of thiamine was added to repress the expression of Slp1 as above.

## Chromatin immunoprecipitation (ChIP)

ChIP was carried out as described previously (Rivosecchi et al, 2019), using the primers listed in Table S2. GFP-tagged proteins were immunoprecipitated with the A11122 antibody (Thermo Fisher Scientific); Myc-tagged proteins were immunoprecipitated with the 9E10 antibody (Merck); Rpb1 was immunoprecipitated using the 8WG16 antibody (Merck).

## Mathematical modelling and simulations (see also the Supplemental Data 1)

We considered a genomic locus of 10 kbp that we modelled as a unidimensional array of $N$ = 100 bins (1 bin = 100 bp). This region contains a 2-kbp-long gene between bin 1 and bin 20. We investigated the interplay between RNAP elongation and condensin translocation using two models.

### *The simple model ([Fig 4A])*

This first model is similar to the one described in Brandão et al (2019). RNAP elongation is modelled as a totally asymmetric simple exclusion process (TASEP) (Derrida et al, 1992; Klumpp & Hwa, 2008; Dobrzynski & Bruggeman, 2009). After binding to the TSS (bin 1) at a rate $\gamma_{init}$, RNAP molecules elongate at a speed rate of $v_{RNAP}$ along the gene from 5′ to 3′ and unbind at a rate $\gamma_{term}$ when they reach the termination zone (bins ≥ 20). Binding at TSS or translocation to the adjacent bin of one RNAP may occur only if the corresponding bin is not already occupied by another RNAP. Condensin translocation occurs either from 5′ to 3′ (bin 1–100, head-to-tail situation) or from 3′ to 5′ (bin 100 to 1, head-to-head situation). Condensin translocates at a speed rate $v_c$ in the absence of adjacent RNAPs in the direction of movement. In the presence of an RNAP, the rate of translocation is reduced to $v_{jump}$. As in Brandão et al (2019), we posited that RNAP translocation is not affected by condensin but that condensin may be pushed towards the 3′ by an adjacent translocating RNAP. For simplicity, we assumed that only one condensin is travelling at a time along the region.

### *The backtrack model ([Fig 4D])*

This second model accounts for the backtracking of RNAP and its impact on condensin translocation. RNAP dynamics is modelled as a TASEP with pauses (Klumpp, 2011; Wang et al, 2014). RNAPs switch between two states: a mobile, elongating state (mRNAP) and a paused, backtracked state (bRNAP). Initiation, elongation, and termination occur as in the simple model except that only mRNAPs can move and that only bRNAPs can unbind. Switching from mobile to paused states happen at rate $k_{back}(i)$ that may depend on the position $i$. bRNAPs become mobile at a homogeneous rate $k_{on}$. Condensin translocates as described in the simple model except that the bypassing rate now depends on the state of the adjacent RNAP ($v_{jump}^m$ and $v_{jump}^b$ for mRNAPs or bRNAPs, respectively).

### *Simulations*

Both models were studied using the standard Gillespie algorithm (Gillespie, 1977) that simulates exact stochastic trajectories for systems of reaction rates. For a given model and parameter set, we simulated $10^5$ different events for each type of condensin translocation (head-to-tail or head-to-head). One event was composed of two steps: (1) a first stage where only RNAP dynamics was simulated to reach a steady-state configuration for RNAPs. RNAP occupancy profiles shown in Fig 4B, F, and H represent the steady-state probabilities to find a RNAP at a given position; (2) a second stage where one condensin is introduced at bin 1 for head-to-tail situations or at bin 100 for head-to-head ones and where the full system is simulated until the condensin reaches bin 100 or bin 1, respectively. During this second stage, we monitored the total time spent by condensin at each bin. Fig 4C, G, and I gives the average of the residence time at a given position over the $10^5$ different simulations.

### *Parameters*

In the simple model (TASEP), we used $v_{RNAP}$ = 40 bp/s (=0.4 bin/s), a typical speed rate for elongating RNAPs (Milo et al, 2010). $\gamma_{init}$ = 0.05 RNAP/s and $\gamma_{term}$ = 0.4 s$^{-1}$ were chosen to obtain a WT-like dense and flat profile for RNAP occupancy. For condensin translocation, we chose $v_c$ = 1 kbp/s, a typical loop extrusion rate observed in vitro (Banigan et al, 2020), and varied $v_{jump}$ from 1 to 15 bp/s. In the backtrack model (TASEP with pauses), we used $v_{RNAP}$ = 40 bp/s, $\gamma_{init}$ = 0.05 RNAP/s, $\gamma_{term}$ = 0.2 s$^{-1}$. $k_{back}(i)$ is given in Fig 4E, assuming a 200-fold stronger rate of backtracking in the termination zone. $k_{on}$ = 0.2 s$^{-1}$ to obtain a flat profile for RNAP occupancy in the WT case and $k_{on}$ = 0.02 s$^{-1}$ for a tilted profile in the *tfs1DN* case (Fig 4H and I). Condensin-related parameters are given by $v_c$ = 1 kbp/s, $v_{jump}^m$ = 15 bp/s and a maximal impact of backtracked RNAPs with $v_{jump}^b$ = 0. A more detailed description of the models can be found in the Supplemental Data 1.

# Data Availability

The in-house MATLAB script developed for simulating the mathematical models from this publication is available at https://github.com/physical-biology-of-chromatin/BackRNAP-Condensin.

# Supplementary Information

# Acknowledgements

We are very grateful to François Bachand (University of Sherbrooke) for sending the *tfs1-DN* over-expression plasmid. This work was supported by a "Chaire d'Excellence" (Project TRACC, CHX11) and by a "Projet de Recherche Collaborative" (PRC) (project 19-CE12-0016-04) awarded to V Vanoosthuyse by the Agence Nationale pour la Recherche (ANR), and by the PRC project (ANR-15-CE12-0002-01) and the project (PJA 20191209370) awarded to P Bernard by the Agence Nationale pour la Recherche (ANR) and the Association pour la Recherche sur le Cancer (ARC), respectively. D Jost acknowledges additional funding by the Agence Nationale pour la Recherche (ANR-18-CE12-0006-03, ANR-18-CE45-0022-01).

## Author Contribution

J Rivosecchi: formal analysis, investigation, and writing—review and editing.

D Jost: conceptualization, formal analysis, investigation, methodology, and writing—review and editing.

L Vachez: investigation.

F Gautier: investigation.

P Bernard: conceptualization, supervision, funding acquisition, and writing—review and editing.

V Vanoosthuyse: conceptualization, formal analysis, funding acquisition, investigation, and writing—original draft, review, and editing.

## Conflict of Interest Statement

The authors declare that they have no conflict of interest.

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
