## [Reviewer comments · Life Science Alliance]

Life Science Alliance

RNA polymerase backtracking results in the accumulation of fission yeast condensin at active genes

Julieta Rivosecchi, Daniel Jost, Laetitia Vachez, François Gautier, Pascal Bernard, and Vincent Vanoosthuyse

DOI: <https://doi.org/10.26508/lsa.202101046>

Corresponding author(s): Vincent Vanoosthuyse, CNRS

Review Timeline:	Submission Date:	2021-02-04
	Editorial Decision:	2021-03-02
	Revision Received:	2021-03-04
	Accepted:	2021-03-17

Scientific Editor: Shachi Bhatt

Transaction Report:

Please note that the manuscript was reviewed at Review Commons and these reports were taken into account in the decision-making process at Life Science Alliance.

Review
COMMONS

Reviewer #1 (Evidence, reproducibility and clarity (Required)):

It is known that condensin complex is accumulated at 3' end of highly transcribed genes in fission yeast mitotic chromosomes. The authors demonstrated that expression of a dominant-negative form of TFIIS (fts1DN), which presumably stabilizes stalled and backtracked RNAP2 throughout a gene body, changed distribution of condensin; its binding at the 3' end of genes was reduced while the binding at the 5' part was increased. This observation was conducted by ChIP-qPCR, at representative four genes (5-7 qPCR sites a gene). Based on this finding, the authors claim that RNAP backtracking plays a key role in positioning of condensin along chromosomes.

****Major comments:****

The key claim is supported only by a single experiment (Fig.3).

We disagree with the referee here. The experiments we carried out in the *sen1Δ* mutant also strongly support our claims. We only detected a significant accumulation of condensin at RNAP3-transcribed genes in the absence of Sen1 and we managed to directly link this accumulation to the concomitant accumulation of RNAP3 over a large domain in the 3' of the gene. As suggested previously (PMID: 31294478), the accumulation of RNAP3 that we detected in the absence of Sen1 is probably due to backtracking (or at least to longer dwell times) because the rates of RNAP3 transcription are actually significantly reduced in this mutant (PMID: 31294478). Our data are therefore consistent with the idea that RNAP3 backtracking over a large domain in the absence of Sen1 results in the accumulation of condensin in -cis. To conclude, both the experiments carried out in the *sen1Δ* and the *fts1DN* mutant conditions are consistent with a role of RNAP backtracking in the positioning of condensin.

Hence, the validity of this experiment should be assessed carefully, in our opinion. The authors used 8WG16 mAb for RNAP2 ChIP, but this mAb is raised for unphosphorylated CTD repeats. Indeed, 8WG16 showed reduced reactivity with phosphorylated RNAP2 (ex. PMID: 11390638). The changed RNAP2 distribution caused by *fts1DN* expression may have been due to altered phosphorylation status of RNAP2. We highly recommend the authors to conduct RNAP2 ChIP-seq with another antibody (with calibration by spike-in control or qPCR conducted in parallel) to visualize the effect of *fts1DN* more accurately. In addition, if the authors can also perform condensin ChIP-seq, it would enable genome-wide and high-resolution analysis of condensin distribution, increasing the impact of Fig 3 greatly.

Our results are consistent with the published literature. As illustrated on Fig 1 and Fig 3, the profiles that we obtained with the 8WG16 antibody are very similar to the profiles obtained by Sutani et al (2015) using a tagged version of Rbp5, which would recognize both phosphorylated and unphosphorylated RNAP2 molecules. As shown by multiple labs, 8WG16 signals tend to accumulate in the 3' end of many fission yeast genes (for example, see PMID: 25240800, PMID: 30341288 & PMID: 30282034), where RNAP2 is heavily hyper-phosphorylated. We therefore believe that, overall, the phosphorylation status of RNAP2 has only a negligible impact on the ability of the 8WG16 antibody to recognize fission yeast RNAP2 molecules and that our experiments captured very well the steady-state levels of RNAP2 along our target genes. In addition, there are several reasons that justify not performing a ChIP experiment with a phosphospecific antibody at this stage:

1. It would only capture a sub-population of RNAP2, which might not be the one impacting condensin,
2. To perform a proper experiment with a phospho-specific antibody would require a mutant control where the target sites cannot be phosphorylated. We know by experience for example that some of the S2P-specific antibodies can still recognize efficiently the CTDS2A mutant, demonstrating that their specificity is not perfect. Some of the unphosphorylatable mutants are lethal and this experiment would therefore be both costly and not trivial to perform.
Finally, as discussed in the manuscript, the impact of tfs1DN over-expression on RNAP2 has already been thoroughly described in different model systems and we feel that another round of ChIP-seq would be slightly redundant with the current literature.

Reviewer #1 (Significance (Required)):

-Nature and significance of the advance:

It remained to be understood how condensin distribution along mitotic chromosomes is determined. The current study addressed this question and provided an interesting clue to it. However, the preferential condensin binding to 3' gene end is only observed in fission yeast. It remains an open question how general and influential the finding in this study is.

We believe that it is not entirely accurate to claim that the preferential binding of condensin in the 3' of genes is only observed in fission yeast. Looking at Figure 1 and Supplementary Figure S1 of d'Ambrosio et al 2008 (PMID: 18708580), it is easy to see a preferential accumulation of condensin in the 3' of genes in budding yeast. Similarly, an accumulation of condensin II in the 3' of active genes has also been reported in vertebrate cells (PMID: 31831736). More importantly, we strongly believe that our study goes beyond the specific case of fission yeast or RNAP2-transcribed genes. We showed that the positioning of condensin is not dictated by a specific transcription termination machinery but instead by the behavior of the RNA polymerase, whether RNAP2 or RNAP3. As noted by referee 2, our data therefore show that the relevant parameter to consider here is the dynamics of the impeding protein rather than its identity. This fits nicely with the recent work of Stéphane Marcand showing that arrays of tightly-bound proteins can interfere with the function of condensin in budding yeast (PMID: 31204167). This latter study did not however evaluate the impact of such tightly-bound proteins on the localization of condensin. Our work therefore both complements and goes beyond this study and firmly establishes arrays of tightly-bound proteins as positioning devices for condensin. This has important consequences for our understanding of the molecular mechanism of condensin translocation.

Reviewer #2 (Evidence, reproducibility and clarity (Required)):

Summary:

This study investigates the mechanisms leading to the enrichment of condensin complexes at the 3'-end of transcribed genes in fission yeast by performing chromatin immunoprecipitation coupled to quantitative PCR. It first shows that gene orientation is not important for condensin localization to a RNAP2-transcribed gene (being consistent with previous findings).

It was shown in prokaryotes that gene orientation strongly impacts the distribution of SMC/condensin but we are not aware of studies that directly tested the impact of gene orientation on the localization of condensin in eukaryotes. As discussed above, we believe that our observation is novel rather than a simple confirmation of previous findings, especially if, as we proposed previously, the loading of condensin is not random but focused at promoter regions of active genes.

Deletion of the Sen1 helicase is shown to promote the accumulation of condensin also to RNAP3-transcribed genes, presumably by causing defective transcription termination of RNAP3. This defect is rescued by strengthening the terminator sequences at two selected genes. The artificial condensin enrichment occurs independent of changes in TFIIC levels, suggesting a transcription-mediated process. Finally, the authors hypothesize that RNAP backtracking at the TTS leads to condensin enrichment. They test this by expressing a mutant TFIIS leading to prolonged backtracking also in gene bodies. As a result, RNAP2 as well as condensin distributions are shifted towards the 5'-end of genes.

Altogether, this study indicates that the state of the transcription apparatus rather than the nature of the involved proteins is relevant for the association with condensin. This is consistent with the idea that RNAPs (possibly multiple tandem RNAPs together) form a barrier for condensin translocation but does not fit well with an alternative scenario of a direct physical interaction between condensin and RNAPs.

The data appear solid and clear. The main conclusions are well supported. The ms is short and concise.

We thank the referee for their positive evaluation of our work.

****Major comments:****

Analysis of some of the ChIP samples by deep sequencing might be useful (at the least to exclude unexpected enrichment occurring elsewhere in the genome and to clarify the generality of the observations).

As determined by the work of Sutani et al (2015), there are only 49 major sites of condensin accumulation on fission yeast chromosome arms. We have therefore looked in details at ~10% of those by ChIP-qPCR. As an alternative to performing expensive and time-consuming calibrated ChIP-seq experiments, which are especially difficult for us to perform at the moment as we only have limited access to the lab due to severe sanitary restrictions in our institute, we have instead decided to complement our work with an in silico modeling approach to evaluate the impact of RNAP backtracking on the distribution of condensin in the vicinity of transcribed regions. We believe that this multi-disciplinary strategy brings a greater benefit to our study.

The physiological relevance of the specific enrichment of condensin at transcribed genes is poorly understood. Can the authors test whether Sen1 deletion or TFIIS mutation leads to delays in chromosome segregation in anaphase (as would be expected for malfunctioning condensin)?

As lack of Sen1 only impacts the association of condensin at specific RNAP3-transcribed genes, we do not anticipate a strong effect on chromosome segregation. Similarly, the over-expression of *tfs1DN* is lethal and any impact on chromosome segregation might be difficult to interpret.

It has previously been proposed that condensin accumulates at ssDNA regions (Sutani et al., 2015). The authors should be able to test whether the correlation holds true for the artificial recruitment observed in this work (Sen1 del, TFIIIS mutant) by pre-treating ChIP samples with nuclease P1 and/or by performing ChIP for Ssb1. This could clarify whether ssDNA is directly involved in condensin localization or not.

As shown at the end of this document, we have tried and failed to reproduce the data presented in Sutani et al. 2015 showing the apparent sensitivity of the condensin ChIP signals to treatment designed to digest single-strand DNA (ssDNA). In our hands and in metaphase-arrested cells at least, our condensin ChIP signals were resistant to increasing amount of Mung Bean or S1 nucleases, whilst such treatments were sufficient to digest control ssDNA (see figure #1 presented below at the end of this document for the referees, p6). This suggests to us that this is not a robust approach to address this question. Similarly, contrary to the Sutani et al 2015 study, we have been unable to detect significant levels of Ssb1 at condensin-binding sites in metaphase-arrested cells (see figure #2 presented below at the end of this document for the referees, p7). More importantly, we believe that the role of ssDNA in the positioning of condensin, if any, should be the focus of another study.

****Minor comments:****

The TFIIIC component Sfc6 (Fig 2B) should be mentioned in the main text.

This has been done.

Fig 3. It would be helpful to display the genes in all panels (left and right) in the same orientation.

As the manner of the presentation of the data has no real impact on their interpretation, we prefer to keep the genes in their chromosomal orientation.

The authors may consider showing EV4 (including the no-tag controls) instead of Fig. 3.

This has been done.

The word 'drives' in the title is maybe too strong.

We have replaced 'drives' by 'results in'.

Introduction: The statement about 'protein scaffold' could be misleading by implying a rigid/solid scaffold.

We have removed this part of the sentence accordingly.

Introduction: 'but the structural details of the formation and enlargement of such loops remain misunderstood' should probably mean '....poorly understood'

This has been changed accordingly.

Introduction: 'The relative contributions of loop extrusion and diffusion capture to chromosome organization remain to be understood.' It is not obvious how this is relevant for the work. Either omit or explain link directly.

We have now deleted this sentence from the introduction.

Reviewer #2 (Significance (Required)):

Condensin localization to transcribed genes has been reported previously and studied in some detail. The mechanisms of condensin localization in general and in particular to the 3'-end of genes, however, are still largely elusive. This study sheds some light on the process.

The observations with Sen1 deletion and over-expression of mutant TFIIIS are novel. The findings indicate that static RNAPs are particularly good barriers for condensin translocation, similar to the tight Rap1-DNA complexes. The authors suggest that more static RNAPs are found (under normal conditions) at TTS. Apart from being more static they might not be otherwise special in condensin recruitment. This hypothesis is new and might be more generally relevant (also for localization of cohesin and other SMC complexes).

The finding that gene orientation is irrelevant for condensin localization is consistent with the barrier model (condensin loading is thought to be wide-spread/random), but not surprising as the accumulation to TTS has been observed for many (thousands of) genes apparently regardless of gene orientation (Sutani et al., 2015).

As explained above, we disagree slightly with the referee on that particular point. We would also like to point out that according to the data by Sutani et al. (2015), condensin only accumulate at ~340 sites in fission yeast, of which only 49 sites are really significant. The study by Sutani et al did not consider the possible role of gene orientation.

Overall, the work reports interesting observations and raises a new hypothesis that may help to guide future research in the right directions.

We thank the referee for their positive and comprehensive appraisal of the significance of our observations.

Reviewer #3 (Evidence, reproducibility and clarity (Required)):

In this manuscript, the authors examine the role of transcription in localising condensin in fission yeast. It had been previously shown that condensin is concentrated at the 3'ends of genes in fission yeast and there was circumstantial evidence in this organism and others of interplay between condensin localization and transcription. The authors present three experiments that strengthen this link and propose a model whereby RNA polymerase backtracking could provide an impediment to condensin translocation. First, they show that changing the orientation of two genes, still leads to condensin accumulation at the 3' end, implying a feature intrinsic to this region in accumulating condensin. Second, the authors impair transcriptional termination using a mutation in *sen1*, and find that a commensurate increase in condensin at transcriptional termini. Interestingly, addition of an artificial terminator rescues both effects. Third, using a mutation known to increase RNAP2 backtracking, the authors find that both RNAP2 and condensin accumulate more towards the 5' end of the genes. Overall, the authors provide further evidence that RNAP2 positions condensin, and suggest a model whereby RNAP backtracking plays a key role in this process.

The presented data are convincing and, for the most part, well-controlled. Although the manuscript does not provide mechanistic insight into the manner by which transcription/RNAP might position condensin, it strengthens the idea that there is a direct causal role of RNAP in condensin localization. Importantly, the authors interpret their data and, while they propose an interesting model based on RNAP back-tracking, they are careful to point out that they are unable to make this definitive conclusion from their data.

We thank the referee for their positive evaluation of our work.

****Major comments****

The main question that arises from this work is whether this is really a specific effect of the act of transcription or whether any DNA-associated protein or machine could do the same. The experiments with LacI are not very compelling in this regard for two main reasons. First, LacI is much smaller than an elongating polymerase. Second, it is not clear that LacI will be enriched on chromosomes to any extent without lacO to bind to. Could the authors target dCas9 or some other DNA-associated protein to address this point more fully? If they choose not to do this experiment, they should tone down their conclusions about the LacI experiment.

We are not sure that we understand the point the referee is making here. We only use GFP-tagged LacI as a specificity control in our experiments to demonstrate that lack of Sen1 did not result in the accumulation of a random, nucleoplasmic but not DNA-bound, heterologous protein, in the 3' of RNAP3-transcribed genes.

****Minor comments****

Another question, though beyond the scope of the current study is whether the different localizations of condensin represent its active movement by loop extrusion or otherwise, loading and re-loading, or some other type of movement. The authors could discuss this.

We have strengthened our Discussion on that point.

Reviewer #3 (Significance (Required)):

Though largely confirmatory, this work includes some elegant and important experiments (such as changing the direction of the gene, use of synthetic terminators) to make clear conclusions. It will be of interest to researchers in the SMC protein field (with which I am familiar) and transcription fields.

We again disagree that our study is merely confirmatory. We believe that our work makes a significant contribution to the understanding of the role of transcription in the positioning of condensin and more largely to other SMC complexes.

changing the direction of the gene, use of synthetic terminators) to make clear conclusions. It will be of interest to researchers in the SMC protein field (with which I am familiar) and transcription fields.

We again disagree that our study is merely confirmatory. We believe that our work makes a significant contribution to the understanding of the role of transcription in the positioning of condensin and more largely to other SMC complexes.

 Addition figures for the referees

Figure 1 for referees: The condensin ChIP signals are resistant to ssDNA-specific nuclease treatments in metaphase cells. Fission yeast cells expressing a GFP-tagged version of the condensin sub-unit Cnd2 were synchronized in metaphase using the *slp1* depletion strain and the distribution of Cnd2-GFP was determined by ChIP. The immuno-precipitated complexes were treated with increasing concentrations of Mung Bean nuclease (A) or Nuclease S1 (B). Heat-denatured genomic DNA from budding yeast was added as control for the digest (see right-hand panel in A). The position of the primers within the genes tested is indicated in the right-hand panel in B: #1 corresponds to the 5', #2 to the gene body and #3 to the termination site.

Figure 2 for referees: The ssDNA-binding protein Ssb1 does not particularly accumulate at condensin-binding sites in metaphase-arrested cells. Fission yeast cells expressing a GFP-tagged version of either the condensin sub-unit Cnd2 or the ssDNA-binding protein Ssb1 were synchronized in metaphase using the *slp1* depletion strain and the distribution of Cnd2-GFP or Ssb1-GFP was determined by ChIP. The position of the primers within the genes tested is indicated in the right-hand panel in **B**: #1 corresponds to the 5', #2 to the gene body and #3 to the termination site (average \pm std of three biological replicates).

March 2, 2021

RE: Life Science Alliance Manuscript #LSA-2021-01046-T

Vincent Vanoosthuyse
Architecture et Dynamique Fonctionnelle des Chromosomes UMR5239 CNRS/ENS-Lyon/UCBL
Laboratoire de Biologie Moléculaire de la Cellule Ecole Normale Supérieure de Lyon
Lyon 69364 Lyon Cedex 07
France

Dear Dr. Vanoosthuyse,

Thank you for submitting your revised manuscript entitled "RNA polymerase backtracking results in the accumulation of fission yeast condensin at active genes". Please accept our sincerest apologies for this extensive delay in getting our decision back to you.

We are happy to inform you that we would like to publish your paper in Life Science Alliance pending final revisions. For a brief background, the manuscript was originally submitted to Review Commons, and then was transferred to Life Science Alliance (LSA) along with the reviewers' comments and a point-by-point rebuttal. LSA editors assessed the submitted documents and deemed the authors' response to the reviewers' concerns appropriate. LSA would like to offer the authors the opportunity to publish the manuscript with us provided appropriate revisions are made in accordance to what is laid out in the pbp rebuttal and to meet our formatting guidelines.

Along with the points listed below, please also attend to the following,

- please make sure the author order in your manuscript and our system match (there is listed Dr. Daniel Jost in the system, but not in the manuscript file)
- please consult our manuscript preparation guidelines <https://www.life-science-alliance.org/manuscript-prep> and make sure your manuscript sections are in the correct order
- please separate the Results and Discussion section into two - 1. Results 2. Discussion, as per our formatting requirements
- please add a Category, a Running Title, a Summary Blurb/Alternate Abstract, a conflict of interest statement, and Author Contribution for your manuscript in our system
- please upload your main and supplementary figures as single files
- please add a callout for Figure EV4 (S4) to your main manuscript text
- please add your main, supplementary figure and table legends to the main manuscript text after the references section
- LSA allows supplementary figures, but no EV Figures; please update your callouts for the Supplementary Figures in the manuscript Fig EV1=Fig S1; while supplementary figures use the system supplementary Fig S1;
- please upload your Tables in editable .doc or excel format, and rename them as well as Supplementary Tables- both in their titles and in their callouts in the manuscript text;
- please upload your main manuscript text as an editable doc file
- please use the [10 author names, et al.] format in your references (i.e. limit the author names to the first 10)
- please label the supplementary figures containing the raw data for main figures as Source Data

files instead of EV files

A. FINAL FILES:

B. MANUSCRIPT ORGANIZATION AND FORMATTING:

Sincerely,

Shachi Bhatt, Ph.D.
Executive Editor
Life Science Alliance
<https://www.lsjournal.org/>
Tweet @SciBhatt @LSAJournal

Interested in an editorial career? EMBO Solutions is hiring a Scientific Editor to join the international Life Science Alliance team. Find out more here -
https://www.embo.org/documents/jobs/Vacancy_Notice_Scientific_editor_LSA.pdf

March 17, 2021

RE: Life Science Alliance Manuscript #LSA-2021-01046-TR

Dr. Vincent Vanoosthuyse

CNRS

Architecture et Dynamique Fonctionnelle des Chromosomes UMR5239 CNRS/ENS-Lyon/UCBL

Laboratoire de Biologie Moléculaire de la Cellule Ecole Normale Supérieure de Lyon

46 allée d'Italie

Lyon 69364 Lyon Cedex 07

France

Dear Dr. Vanoosthuyse,

Thank you for submitting your Research Article entitled "RNA polymerase backtracking results in the accumulation of fission yeast condensin at active genes". It is a pleasure to let you know that your manuscript is now accepted for publication in Life Science Alliance (LSA). Congratulations on this interesting work.

We understand that you have not been able to perform the in silico modeling to evaluate the impact of RNAP backtracking on the distribution of condensin in the vicinity of transcribed regions that you had proposed to include in response to Reviewer 1's concerns. Given the reviewers' overall positive outlook on the manuscript and the fact that the conclusions are well supported by the data, we have decided to over-rule this request for LSA.

*****IMPORTANT:** If you will be unreachable at any time, please provide us with the email address of an alternate author. Failure to respond to routine queries may lead to unavoidable delays in publication.*******

DISTRIBUTION OF MATERIALS:

Authors are required to distribute freely any materials used in experiments published in Life Science Alliance. Authors are encouraged to deposit materials used in their studies to the appropriate

repositories for distribution to researchers.

Again, congratulations on a very nice paper. I hope you found the review process to be constructive and are pleased with how the manuscript was handled editorially. We look forward to future exciting submissions from your lab.

Sincerely,

Shachi Bhatt, Ph.D.

Executive Editor

Life Science Alliance

<https://www.lsjournal.org/>

Tweet @SciBhatt @LSAJournal